# Synthetic virions reveal fatty acid-coupled adaptive immunogenicity of SARS-CoV-2 spike glycoprotein

Oskar Staufer [1,2,3,4✉], Kapil Gupta [5,6], Jochen Estebano Hernandez Bücher[1,2], Fabian Kohler[7], Christian Sigl[7], Gunjita Singh[5], Kate Vasileiou[5], Ana Yagüe Relimpio[1,2], Meline Macher [1,2,4], Sebastian Fabritz[8], Hendrik Dietz [4,7], Elisabetta Ada Cavalcanti Adam [1,4], Christiane Schaffitzel [5,6,9], Alessia Ruggieri [10], Ilia Platzman[1,2,3], Imre Berger [3,5,6,9✉] & Joachim P. Spatz [1,2,3,4✉]

SARS-CoV-2 infection is a major global public health concern with incompletely understood pathogenesis. The SARS-CoV-2 spike (S) glycoprotein comprises a highly conserved free fatty acid binding pocket (FABP) with unknown function and evolutionary selection advantage[1,2]. Deciphering FABP impact on COVID-19 progression is challenged by the heterogenous nature and large molecular variability of live virus. Here we create synthetic minimal virions (MiniVs) of wild-type and mutant SARS-CoV-2 with precise molecular composition and programmable complexity by bottom-up assembly. MiniV-based systematic assessment of S free fatty acid (FFA) binding reveals that FABP functions as an allosteric regulatory site enabling adaptation of SARS-CoV-2 immunogenicity to inflammation states via binding of pro-inflammatory FFAs. This is achieved by regulation of the S open-to-close equilibrium and the exposure of both, the receptor binding domain (RBD) and the SARS-CoV-2 RGD motif that is responsible for integrin co-receptor engagement. We find that the FDA-approved drugs vitamin K and dexamethasone modulate S-based cell binding in an FABP-like manner. In inflammatory FFA environments, neutralizing immunoglobulins from human convalescent COVID-19 donors lose neutralization activity. Empowered by our MiniV technology, we suggest a conserved mechanism by which SARS-CoV-2 dynamically couples its immunogenicity to the host immune response.

[1] Department for Cellular Biophysics, Max Planck Institute for Medical Research, Jahnstraße 29, 69120 Heidelberg, Germany. [2] Institute for Molecular Systems Engineering, University of Heidelberg, Im Neuenheimer Feld 253, 69120 Heidelberg, Germany. [3] Max Planck-Bristol Center for Minimal Biology, University of Bristol, 1 Tankard's Close, Bristol BS8 1TD, UK. [4] Max Planck School Matter to Life, Jahnstraße 29, 69120 Heidelberg, Germany. [5] School of Biochemistry, Biomedical Sciences, University of Bristol, 1 Tankard's Close, Bristol BS8 1TD, UK. [6] Bristol Synthetic Biology Centre BrisSynBio, University of Bristol, 4 Tyndall Ave, Bristol BS8 1TQ, UK. [7] Department of Physics, Technical University of Munich, 85748 Garching, Germany. [8] Department for Chemical Biology, Max Planck Institute for Medical Research, Jahnstraße 29, 69120 Heidelberg, Germany. [9] Halo Therapeutics Ltd, Science Creates, Albert Road St. Philips Central, Bristol BS2 0XJ, UK. [10] Department of Infectious Diseases, Molecular Virology, Center for Integrated Infectious Disease Research, University of Heidelberg, Im Neuenheimer Feld 344, 69120 Heidelberg, Germany. ✉email: oskar.staufer@mr.mpg.de; imre.berger@bristol.ac.uk; spatz@mr.mpg.de

The global COVID-19 pandemic, caused by severe acute respiratory syndrome–coronavirus 2 (SARS-CoV-2), continues to damage communities and economies, urgently requiring effective antiviral strategies. SARS-CoV-2 infection is initiated by binding of the virus to the target receptor angiotensin-converting enzyme 2 (ACE2), mediated by the trimeric S glycoprotein[3]. Initial binding is achieved by the RBD within the S1 portion of S, while S2 induces fusion of the viral envelop with the target cell membrane[4]. S is the major focus of COVID-19 vaccine development, the primary target for neutralizing antibodies[5] and the main diagnostic antigen for SARS-CoV-2 infections[6]. Structural analyses of S revealed substantial conformational rearrangements of RBD during the infection process[4,7,8]. In a closed conformation, the RBDs lie flat on the S surface and are therefore less accessible for ACE2 binding[9]. In open conformations, at least one RBD erects from S to expose the receptor-binding motif, enabling ACE2 binding. Each monomer within the S trimer can undergo a closed-to-open transition and consecutive priming by ACE2[4]. On intact SARS-CoV-2 virions, ~31% of S trimers are found in a closed state, ~55% with one open RBD and ~14% with two open RBDs[10]. Molecular mechanisms that modulate S open-to-closed equilibrium and their implications for SARS-CoV-2 infection and immune evasion remain poorly understood. Exogenous control of this equilibrium, however, could greatly benefit COVID-19 therapy development and promote prospective pandemic prevention.

Our previous cryogenic electron microscopy (cryo-EM) structure of S revealed a free fatty acid-binding pocket (FABP) within the RBD, tightly binding linoleic acid (LA)[1]. Lined by hydrophobic residues, the RBD forms a kinked "greasy" tube that engulfs the hydrophobic free fatty acid (FFA) hydrocarbon chain of LA. The hydrophilic FFA carboxyl group is anchored by arginine (Arg[408]) and glutamine (Gln[409]) residues from an adjacent RBD in the trimer, forming a polar lid. The FABP is conserved among pathogenic human corona viruses (hCoV) SARS-CoV, SARS-CoV-2, and the Middle East Respiratory Syndrome Coronavirus (MERS-CoV). Importantly, purified S adopts predominantly a locked conformation when LA is bound[1], indicating that FFAs can modulate the S conformational equilibrium potentially impacting infectivity. However, the inherent function of FABP, its impact on COVID-19 pathogenesis and its role for SARS-CoV-2-host adaptation are currently elusive. Interestingly, wide-ranging FFA profile remodeling has been reported in hCoV patients[11–13] and supplementation of FFAs could significantly ameliorate clinical outcomes of SARS-CoV-2 infection[14,15]. In this context, FFAs function as eicosanoids precursors and are key inflammatory regulators in COVID-19 pathogenesis[16,17]. Linking the pathophysiological FFA conditions in COVID-19 to S function as well as a systematic analysis of FABP-based effects in this context have been impeded by the structural complexity of the S protein as well as the undefined nature and low experimental control over SARS-CoV-2 live virions.

In this work, to elucidate the function of FABP and its impact on COVID-19 disease progression, we develop a bottom-up approach for assembling MiniVs as defined, low-complexity synthetic SARS-CoV-2-like particles. A major advantage of our synthetic virions over live SARS-CoV-2 and pseudoviruses is the ability to test S binding exercising precise control over the FFA-loading in the FABP in a safety level 1 laboratory environment. This enables a systematic assessment of FABP-based S functional regulation of SARS-CoV-2 infectivity and IgG immunogenicity.

## Results

### Bottom-up assembly of minimal SARS-CoV-2 virions.
To assess FABP-regulated SARS-CoV-2 receptor-binding, we designed and implemented a bottom-up approach for in vitro assembly of liposome-based synthetic SARS-CoV-2 MiniVs with reconstituted SARS-CoV-2 S ectodomains (Fig. 1a). This approach allows for a systematic analysis of cell attachment effects with molecularly defined virus-like particles. In contrast to pseudovirus approaches and in vitro protein binding assays, our synthetic MiniVs embody the biophysical features of SARS-CoV-2 virions, allowing for integrated assessment of particulate and multivalent binding effects[18]. Moreover, as MiniVs lack genetically encoded information, their interactions with living cells can be investigated under low-biosafety constraints. We produced MiniVs from small unilamellar vesicles (SUVs) designed to mimic the most critical biophysical features involved in SARS-CoV-2 binding. Lipid formulations were adjusted to resemble the virion lipid membrane formed from the endoplasmic-reticulum-Golgi intermediate compartment (ERGIC) (45 mol% DOPC, 21 mol% DOPE, 3 mol% DOPS, 12 mol% DOPI, 14 mol% cholesterol, 3 mol% SM) (Fig. 1b)[19]. We applied MiniVs with a diameter of 112 nm ($\pm$31 nm), matching the SARS-CoV-2 virion size (Fig. 1c)[20]. MiniVs were decorated with recombinant SARS-CoV-2 D614GS (C1-G1236 with C-terminal trimerization domain and histidine tag) via NTA(Ni$^{2+}$) functionalized lipids. Consistent with natural SARS-CoV-2 virions, MiniVs had a negative zeta potential of $-39.2$ mV in H$_2$O and $-3.4$ mV in isosmotic buffer[21]. S immobilization and MiniV ultrastructure was verified by cryo-EM tomographic imaging (Fig. 1d, Fig. S1 and Movie S1). On average, we counted 14.6 ($\pm$5.4, $n = 5$ vesicles) spikes per vesicle, which is comparable to the trimer density found on live SARS-CoV-2 virions[10]. These immobilized, prefusion-stabilized S variants with inactivated polybasic cleavage sites were applied to specifically analyze RBD-mediated receptor-binding independently from S2-induced membrane fusion. MiniV functionality and binding specificity towards ACE2 receptors were verified by quartz crystal microbalance with dissipation monitoring (QCM-D). We found that MiniVs specifically bind to ACE2 receptors with neglectable non-specific attachment to lipid membranes (see Fig. S2). Taken together, our SARS-CoV-2 MiniVs accurately mimic the structure and receptor-binding of natural SARS-CoV-2 virions.

We next incubated MiniVs with ACE2-expressing human epithelial MCF-7 cells[22], A549 human alveolar basal epithelial cells and primary human umbilical vein endothelial cells (HUVEC). Interactions between MiniVs and cells were assessed by confocal microscopy via imaging fluorescent rhodamine lipids integrated into the MiniV membrane. Following the viral particle concentration found in the sputum and the upper respiratory track during SARS-CoV-2 infection[20], we applied $2.4 \times 10^8$ MiniVs mL$^{-1}$ as measured by nanoparticle tracking analysis. Initial single MiniV binding to the target cell membrane was observed after only 10 min (Fig. 1e and Movie S2). Longer incubation for several hours resulted in excessive binding of the MiniVs to the cells (Fig. 1f and Fig. S3). Intriguingly, in A549 epithelial and HUVEC cells, we observed noticeable intracellular uptake of the MiniVs. To quantify S-mediated binding of MiniVs over time in high throughputs, we developed retention assays based on quantification of MiniV fluorescence (see methods) and compared MiniV retention to naive SUVs, lacking S on the surface (Fig. 1g–i). S significantly increased vesicle binding over the observation time period. These MiniV interaction kinetics are in accordance with previous findings for natural hCoV virions[23]. Increased binding was also observed for MiniVs presenting only a recombinant S1 domain (V16-R685) and with S ectodomains of other pathogenic hCoVs, e.g., SARS-CoV and MERS-CoV (Fig. 1j). We further performed competition assays between SARS-CoV-2 viruses and MiniVs. ACE2-expressing A549 human lung epithelial cells[24] were pre-incubated with MiniVs and SARS-CoV-2 infection efficiency was measured by qRT-PCR amplification of Orf7a mRNA. We found that MiniVs are able to competitively block SARS-CoV-2 infection, underscoring the similarities in cell and receptor tropism between MiniVs and

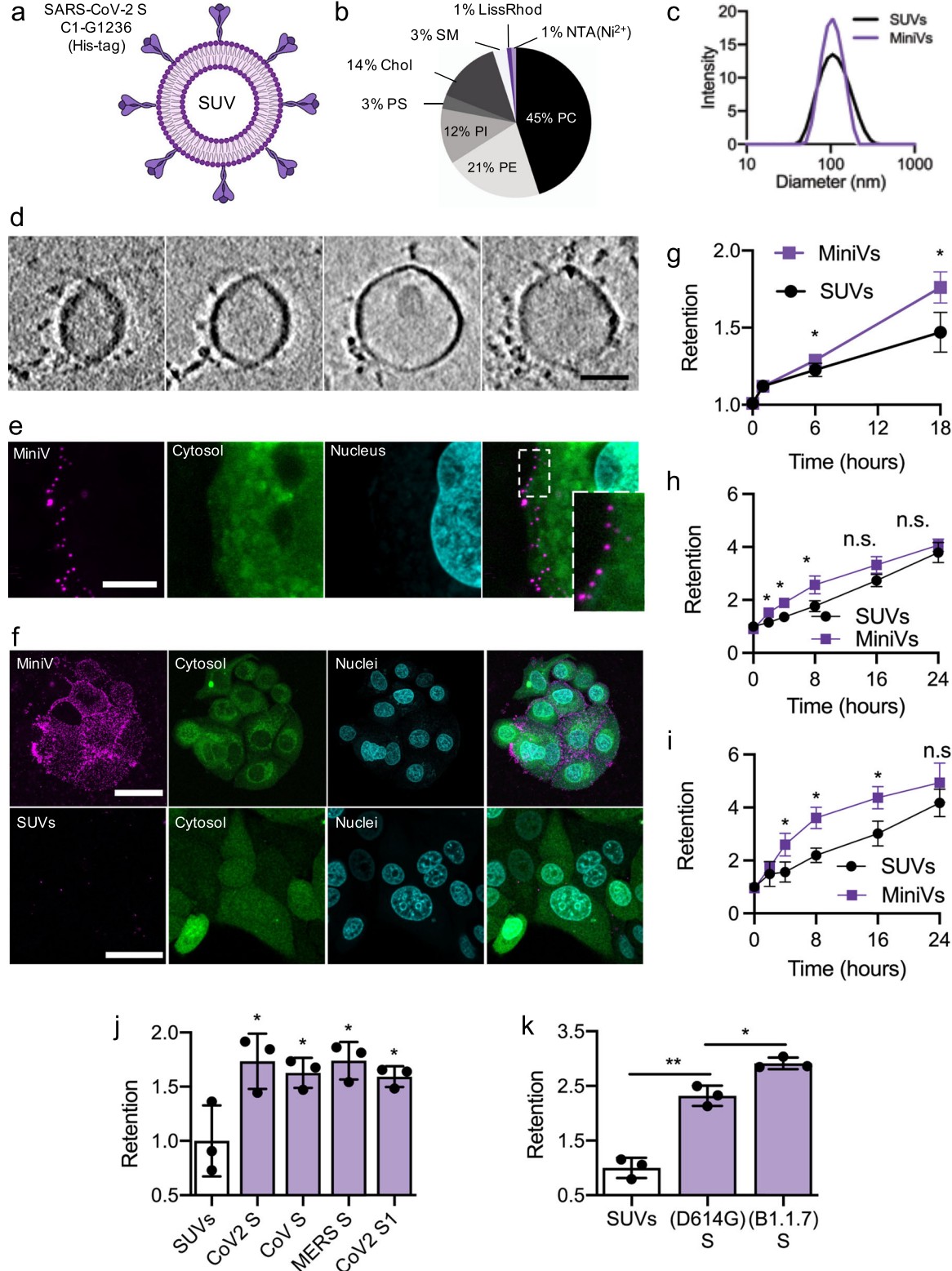

SARS-CoV-2 virions (see Fig. S4). Comparing the retention of D614GS and S variants of the newly emerged B1.1.7 Variant of Concern (UK or alpha mutant), we observed increased binding for B1.1.7 MiniVs, which is in accordance with higher ACE2 affinity[25] (Fig. 1k). Moreover, confocal microscopy assessment showed increased multiplicity of attachment for B1.1.7 MiniVs (see Fig. S5). Taken together, MiniVs mimic the binding of natural SARS-CoV-2 viruses to target cells, faithfully simulating an early event in

COVID-19. Moreover, MiniVs represent a modular approach to adaptively mimic and study different hCoV S mutational variants and strains. Therefore, we applied MiniVs to systematically study FABP-regulated events in initial cell-binding.

**FABP regulates S to cell-binding**. MiniVs allowed for a systematic assessment of changes in S-mediated cell interactions as a

**Fig. 1 Bottom-up assembly of minimal SARS-CoV-2 virions. a** Schematic illustration of MiniVs based on SUVs with SARS-CoV-2 S ectodomains, immobilized via their His-tag. **b** Lipid formulation of SUVs derived from the ERGIC with NTA-functionalized and fluorescent lipids. **c** MiniV and SUV size distribution analysis by dynamic light scattering. **d** Exemplary cryo-EM tomography slices of MiniVs with immobilized S on the membrane. Scale bar is 50 nm. **e** Representative confocal microscopy images, from two independent experiments, of MCF7 human epithelial cells incubated for 10 min with MiniVs, showing attachment of the MiniVs to the cells surface. Inset shows magnified area of attachment. Scale bar is 7 µm. **f** Maximal confocal microscopy z-projections of MCF7 human epithelial cells incubated for 18 h with MiniVs (top row) or with SUVs lacking S on the surface (bottom row). Scale bar is 40 µm. **g–i** Time-resolved retention assay of MiniVs and SUVs incubated with MCF7 (**g**), A549 (**h**), and HUVEC (**i**) cells. **j** SUV-normalized retention assay for MiniVs presenting different recombinant hCoV S variants incubated for 24 h with MCF7 cells. **k** Retention assay for MiniVs presenting SARS-CoV-2 D614G and B1.1.7 S variants incubated for 24 h with MCF7 human epithelial cells. Results in **g–k** are shown as mean ± SD from at least $n = 3$ biological replicates in each experimental condition, *$p < 0.05$, **$p < 0.005$, unpaired two-tailed $t$-test. Source data are provided as a Source Data file.

function of FABP occupancy. Because recombinant native S binds an a priori undefined set of FFAs during expression and purification, we first generated FFA-depleted S (ApoS) by treatment of the purified native S samples with alkoxylated lipophilic columns (see Methods and Table S1)[1]. Compared to the native S, ApoS-decorated MiniVs displayed increased binding to human epithelial cells (Fig. 2a). This is consistent with a model where FFAs lock a closed RBD conformation, thereby reducing exposure of the receptor-binding motif (RBM). We then performed controlled loading of ApoS-MiniVs by incubating them with FFAs of differing length and saturation (i.e., palmitic acid (PA), oleic acid (OA), LA, and arachidonic acid (AA)). MiniVs were loaded with 2× molar excess of FFAs with respect to FABPs in S, equaling 1 µM, which is comparable to the FA levels in sera of COVID-19 patients[26,27]. Based on the nanomolar affinity of S for FFAs[1], this concentration is expected to result in complete saturation of all binding pockets as partially supported by our mass spectrometry quantification (see below). Our choice of FFAs was based on an analysis of the most prominent changes in lipidomic profiles during COVID-19 infection[12]. The binding of OA, LA, and AA, but not PA, in S was successfully verified by multiple reactions monitoring LCMS/MS (see Table S1). Of note, the addition of saturated PA did not significantly reduce MiniV-cell-binding compared to ApoS. However, we found that the (poly)-unsaturated FFAs OA, LA and AA were able to reduce MiniV binding compared to ApoS-MiniVs (Fig. 2a–c and Fig. S6). This is consistent with the FABP tube-like structure that features a kinked hydrophobic pocket accommodating cis-bond unsaturated FFAs and our MS analysis of FFA-binding by S. Importantly, when applying MiniVs presenting only the S1 domain without trimerization site, incubation with polyunsaturated FFAs did not significantly impact on cell-binding (see Fig. S7a). This is in agreement with the FABP structure, where a second adjacent RBD in the trimer is required to stabilize FFA-binding by coordinating the hydrophilic carboxy-group[1]. Treatment of cells with 1 µM soluble FFAs alone, did not significantly decrease the binding of SUVs, demonstrating that FFAs directly act on S-mediated receptor interactions and that changes in MiniV binding are not based on cellular phenotypic alterations (see Fig. S7b). Moreover, loading of MiniVs with FFAs did not substantially change their zeta potential, indicating no alterations in charge-based cell-MiniV interactions due to incorporation of the negatively charged FFAs into the SUV bilayer (see Fig. S7c). Upon initial S-mediated binding of SARS-CoV-2 to the target cell, S is cleaved by cellular membrane proteases (e.g., TMPRSS2) within the S2 domain[28]. This induces fusion of the viral envelop with the target cell membrane, a process that is blocked by the serin protease inhibitor camostat mesylate[29]. Interestingly, we did not measure any significant difference in the retention of FFA-loaded MiniVs upon camostat mesylate incubation (see Fig. S8). This indicates that the FABP is most crucial for the regulation of initial cell-binding rather than for S post-translation modification, fusogenic transformation and envelop fusion. Taken together,

these results demonstrate that the FABP impacts S-mediated cell-binding, presumably via changes in the open-to-closed RBD equilibrium.

SARS-CoV-2 has 76% sequence identity to SARS-CoV[30] and significantly higher infectivity, which is attributed to increased ACE2 binding affinity[31], more elaborate S posttranslational processing[32] and newly emerged cell entry mechanisms[33]. Among others, SARS-CoV-2 acquired a K403R mutation that introduces a RGD motif into the RBD nested next to the receptor-binding motif (Fig. 2d). This motif has been suggested to recruit cell surface integrins as co-receptors, potentially contributing to the increased infectivity compared to SARS-CoV[33]. In our locked LA-bound S structure, the RGD motif is located above the FFA hydrophilic head supporting arginine residue, suggesting a FABP-regulated exposure (Fig. 2e). In order to assess the functional contribution of the RGD motif, we produced a SARS-CoV-2 S(R403A). We found that S(R403A)-presenting MiniVs displayed significantly reduced cell-binding compared to native S MiniVs (Fig. 2f). This suggests that the RGD motif directly contributes to S-based cell-binding. To further verify a potential contribution of the RGD motif, we performed integrin blocking experiments with linear RGD (linRGD) peptides (see Fig. S9a, b). Incubation of native S MiniVs with 20 µM linRGD reduced binding to levels comparable to S(R403A) MiniVs. Importantly, integrin blocking did not affect the binding of MiniVs presenting SARS-CoV S (see Fig. S9c). This indicates that SARS-CoV-2 S can engage integrins for cell entry, although the RGD motif is located proximal to the receptor-binding motif, suggesting a more sequential, rather than simultaneous binding of integrins and ACE2 (see Fig. S10) as observed for other enveloped virus (e.g., integrin $\beta_1$ for mammalian reovirus[34]). To assess if the FABP also regulates S-integrin binding, we measured changes in the binding efficiency of FFA-loaded ApoS-MiniVs blocked with linRGD (Fig. 2g). For this, we measured the native S-normalized differences in retention between linRGD-blocked cultures and control cultures incubated with MiniV that were loaded with FAs (see methods for normalization of retention). As suggested by our cryo-EM structure, we found that unsaturated FFAs influence the engagement of integrins by S, most likely by modulation of the open-to-close RBD equilibrium. Taken together, this demonstrates that the FABP is also able to regulate RGD exposure and enhance integrin engagement by SARS-CoV-2 S.

In the search for therapeutic FABP antagonists that inhibit SARS-CoV-2 receptor-binding, a library of FDA-approved drugs has been screened by molecular simulations[35]. These calculations of FABP binding energy suggested compounds with structural similarity to FFAs as FABP ligands (see Fig. S11). Towards an experimental assessment of these predictions and potential pharmacologic modulation FABP activity, we measured drug-normalized retention (see methods for normalization approach) of ApoS-MiniVs during 1 µM treatment with five of these compounds (Fig. 2h). We found two drugs, vitamin K and dexamethasone, that reduce S-mediated binding of the MiniVs in

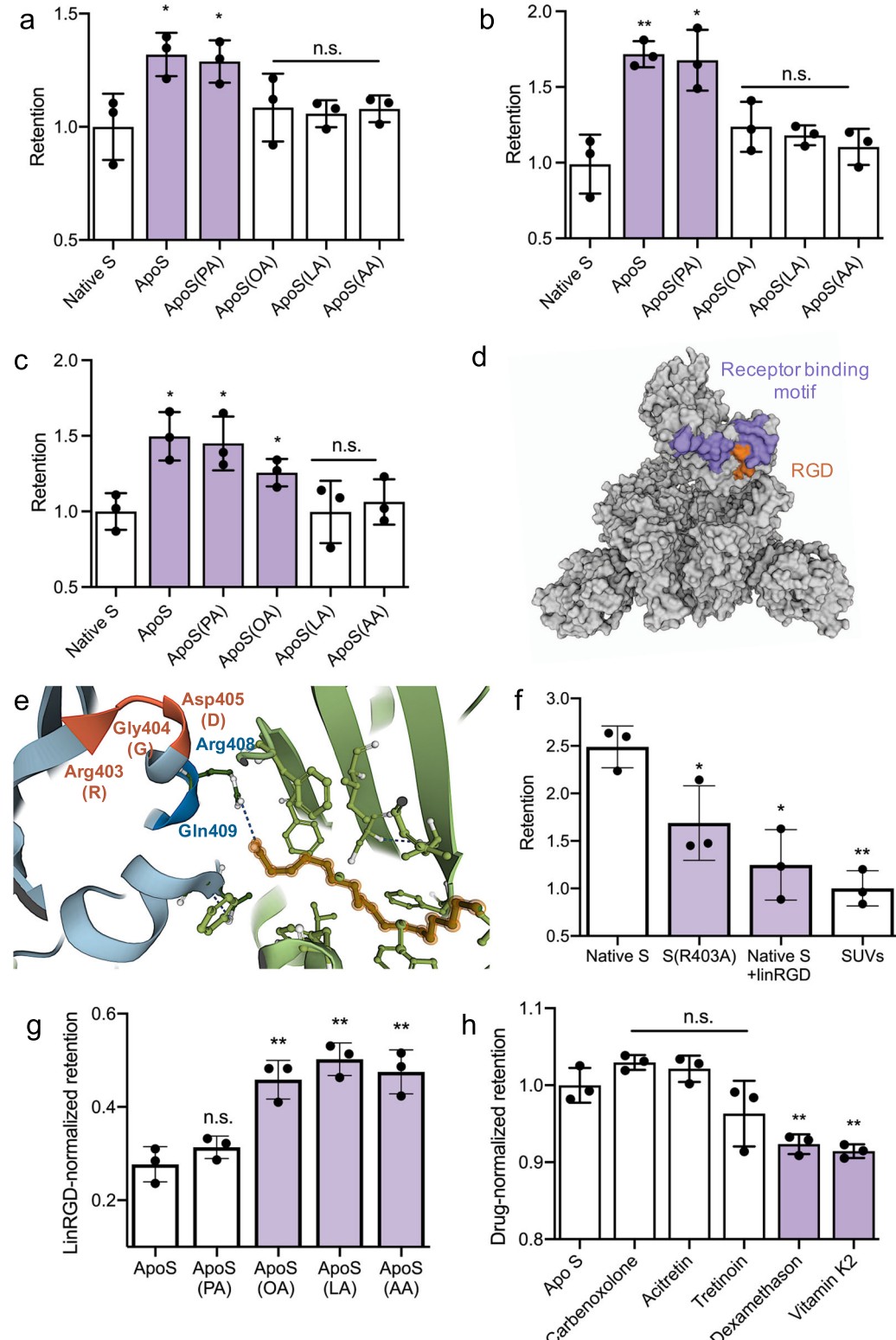

a FABP-regulated manner. We found an IC$_{50}$ concentration of 3.2 μM and 4.7 μM for MiniV retention in A549 human alveolar basal epithelial cells for dexamethasone and vitamin K, respectively. Interestingly, reduced vitamin K levels have previously been identified as a modifiable risk factor of severe COVID-19[36], while dexamethasone is one of a few drugs approved for the treatment of ventilated COVID-19 patients[37]. This corroborates that the FABP is a potentially druggable regulator of SARS-CoV-2 cell-binding.

**FABP regulates S immunogenicity against neutralizing immunoglobulins.** Intriguingly, all key FABP residues are conserved among the globally emerging SARS-CoV-2 Variants of Concern (Fig. 3a) as well as among previous highly pathogenic hCoV[1] and corona viruses found in intermediate species[38]. This persistent structural reoccurrence is remarkable in the light of the fact that FABP apparently restricts rather than facilitates viral cell-binding. A complete absence of FABP disrupting mutations in all highly contagious variants hints at an evolutionary selection

**Fig. 2 FABP-based regulation of S binding. a–c** Native S-normalized retention assay for MiniVs presenting ApoS, loaded with 1 µM FFAs, after incubation with MCF7 cells (**a**), A549 human alveolar basal epithelial cells (**b**), and human umbilical vein endothelial cells (**c**). **d** Molecular surface representation of the S trimer cryo-EM structure (PDB 7BNN) in top view with one open RBD exposing the RGD motif (orange). ACE2 binding residues in the receptor-binding motif are shown in purple. **e** Cartoon structure representation (PDB 6ZB5) of the LA-bound FABP (green), the acidic headgroup anchor (blue), and the adjacent RGD motif (orange) in the LA-locked S conformation. **f** SUV-normalized retention assay for MiniVs presenting native S, R403A mutated S without RGD motif, or native S incubated with 20 µM linRGD for integrin blocking. Retention was measured after 24 h incubation with MCF7 cells. **g** LinRGD-normalized retention assays for MiniVs presenting FFA-loaded ApoS after 24 h incubation with MCF7 cells. **h** Drug-normalized retention assay for ApoS-MiniVs incubated with MCF7 cells and 1 µM of potential S binding drugs. Results **a–c** and **f–h** are shown as mean ± SD from at least $n = 3$ biological replicates in each experimental condition, *$p < 0.05$, **$p < 0.005$, n.s. not significant, unpaired two-tailed $t$-test. Source data are provided as a Source Data file.

advantage provided by FFA-binding. For sustainable infection and viral replication, non-genome-integrating RNA-viruses balance infectivity against immunogenicity during the inflammatory response, e.g., high initial contagiousness in the first days[39] and subsequent transition into an immune-evasive "stealth phase" after the incubation period[40]. In this regard, the RBD is the central immunogenic structure of SARS-CoV-2, accounting for ~90% of IgG neutralizing activity[5,41]. Likely, the FABP regulates conformational changes in S immunogenicity as a function of FFA-binding. We applied MiniVs to systematically evaluate FFA-based changes in S immunogenicity and assessed changes in IgG epitope accessibility between open and closed RBD states. First, we computed the accessible surface area (ASA) of S residues in the open and FFA-locked states (see Fig. S12). Second, we calculated ASA ratios along RBD opening for individual residues to measure changes in the exposure of specific epitopes of neutralizing SARS-CoV-2 IgGs. To verify this strategy, we computed changes in epitope exposure of (i) IgG S2H14, which uniquely neutralizes an open RBD state by competitive blocking of the receptor-binding motif in S[5] (Fig. 3b); (ii) IgG CR3022 that exclusively binds open RBD states and inhibits S-mediated entry by a disruptive mechanism[42,43] (Fig. 3c); and (iii) S2H13 that binds a β-hairpin in the receptor-binding motif accessible in both open and closed states[5] (Fig. 3d). For the CR3022 and S2H14 epitopes (open RBD only binders), we found an average open-to-close ASA ratio of 4.57 and 1.65, respectively. This shows that these IgG epitopes are more accessible in the open state, consistent with their neutralization pattern. In contrast, for the S2H13 epitope (open and closed RBD binder), the average open-to-close ASA ratio was 0.89. This indicates that the S2H13 epitope is even slightly more accessible in the closed conformation, consistent with S2H13 neutralization pattern. This endorses the ASA ratio as an indicator for IgG epitope accessibility in S.

General IgG neutralizing immunodominant sites (NIDS) in S have been identified by mutation mapping from 17 convalescent COVID-19 patient sera[44]. To measure epitope exposure for polyclonal neutralizing human serum IgGs in open and closed states, we calculated the ASA ratio for NIDS in open and closed states (Fig. 3e). We found an average ASA ratio of 1.71, indicating that the key sites for S neutralization are more accessible in the open RBD state, which is in agreement with findings from previous molecular dynamics simulations of S[6]. Importantly, NIDS residues largely overlap with the receptor-binding motif (see Fig. S13a, b). Thus, we raised a VHH nanobody against the receptor-binding motif (ADAH11) in vitro using ribosome display[45] to experimentally validate FABP-influenced neutralization by NIDS-targeting immunoglobulins. ADAH11 efficiently reduced the binding of native S MiniVs (Fig. 3f) with an ED$_{50}$ of 117 nM (see Fig. S13c). Importantly, ADAH11 neutralization was strongly dependent on S loading with FFA, where again only unsaturated FFAs reduced the neutralization efficiency (Fig. 3g and Fig. S14). This is in agreement with reduced neutralization for closed RBD states. In line with this, we found that CR3022

(open RBD only binder) is likewise able to reduce cell-binding of native S MiniVs (Fig. 3h) and CR3022 neutralization activity was sensitive to S loading with unsaturated FFAs (Fig. 3i). Taken together, this suggests that FFA-binding can regulate S neutralization by IgGs by reducing exposure to NIDS.

FFAs, particularly LA and AA, are essential eicosanoid precursors and tissue inflammatory regulators[46]. Basal non-esterified ω-6 FAA levels are maintained below 0.1 µM under physiological conditions[26,46] but can temporarily increase in a so-called FA-lipid storm over seven-fold during hCoV infection, lung inflammation and COVID-19[12,27,47]. To test a possible link between altered FFA levels and S IgG immunogenicity in direct relevance for COVID-19 patients, we assessed MiniV neutralization by IgGs derived from the serum of six convalescent donors under "basal" (0.1 µM) and "inflammatory" (1.1 µM) LA/AA concentrations. We found that patient serum IgGs effectively neutralize MiniV binding under basal LA/AA concentration but lose neutralization activity under elevated LA/FA conditions (Fig. 3j). This reduction in neutralization activity was also found for live SARS-CoV-2 virus infection of human lung epithelial cells, were the IC50 values of the COVID-19 convalescent serum-derived IgG dropped by approximately one order of magnitude (see Fig. S15). This suggests that FFA-binding by S modulates the exposure of immunodominant sites (see Fig. 3e), thereby coupling FFA concentrations to S immunogenicity.

## Discussion

Our study introduces technology for the bottom-up assembly of synthetic SARS-CoV-2-like liposomes (see Fig. 1). MiniVs are modular, adaptive systems that allow quantitative and flexible assessment of different S-variants while precisely controlling the composition and biophysical properties of the particles. As programmable SARS-CoV-2 models, our MiniVs provide a modular framework for COVID-19 research and are particularly attractive as they can be deployed under biosafety level 1 condition. The SARS-CoV-2 MiniVs enabled us to perform systematic analyses of S binding to target cells under conditions exactly defined at the molecular level, in particular with respect to S-mediated cell-binding under FFA-free conditions and with defined FAA profiles as well as with respect to the newly acquired SARS-CoV-2 RGD motif (see Fig. 2e). We found that unsaturated FFA-binding reduced S-mediated cell-binding (see Fig. 2a) and that the RGD motif contributes to the enhanced cell attachment. We further assess the direct impact of FDA-approved drugs as potential FABP ligands on S' cell-binding (see Fig. 2b), which could open up avenues for treating COVID-19 by locking S in a closed conformation.

Our study reveals a mechanism where FFAs function as molecular switches by which SARS-CoV-2 can adapt its immunogenicity to local inflammatory states and host immune response (Fig. 3k). Thus, the FABP represents a dynamic responsive element that provides an evolutionary advantage as it allows for a temporary escape from neutralizing IgG during peak

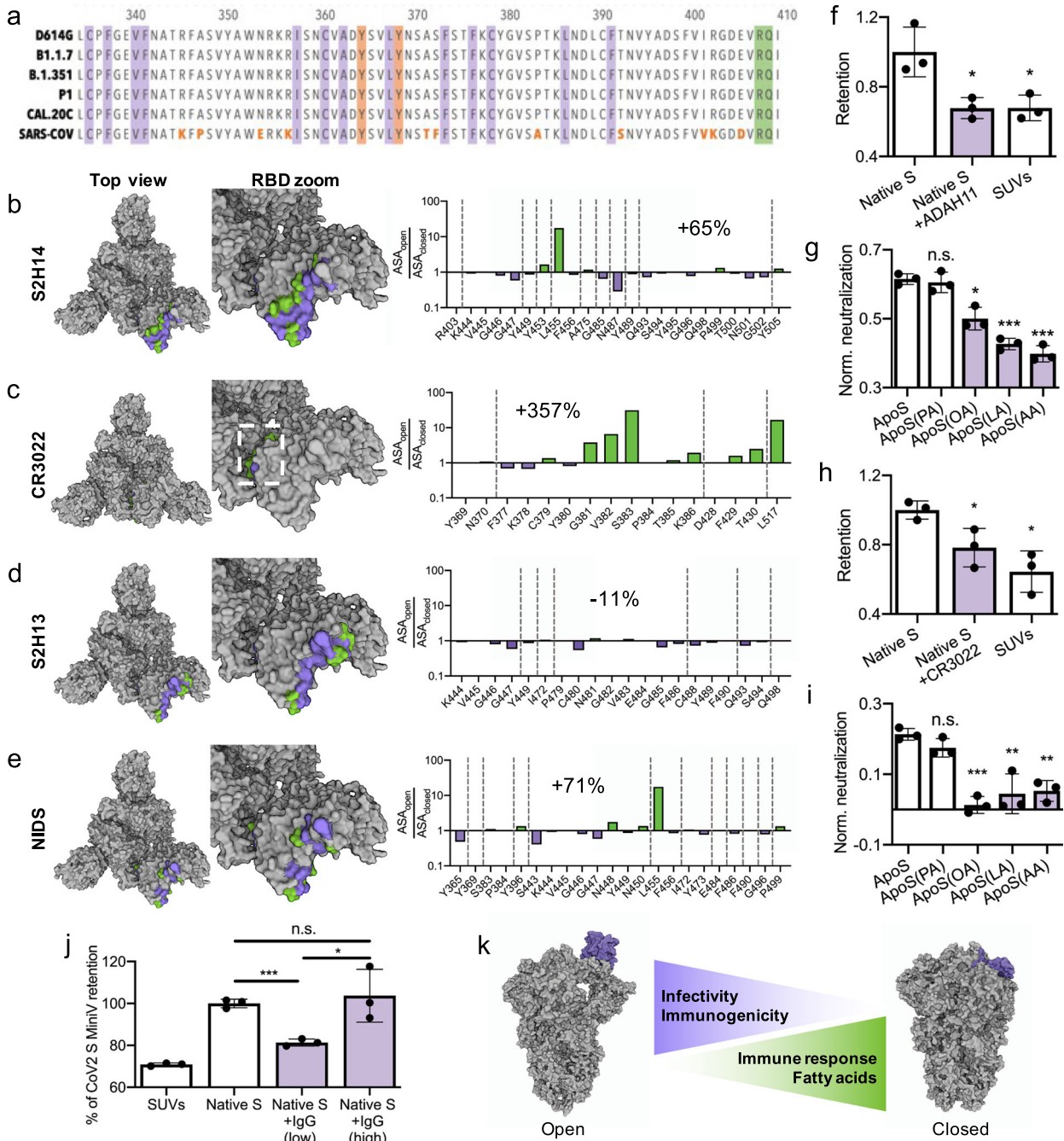

**Fig. 3 FABP-regulated exposure of immunogenic S epitopes. a** Sequence alignment of the FABP from five SARS-CoV-2 variants of concern and SARS-CoV. Residues of the hydrophobic pocket are highlighted in purple, the hydrophilic head-stabilizing residues in green and the gating helix tyrosine in orange. Residues differing in SARS-CoV are written in orange. **b–e** Molecular surface representation of open S (7BNN) with ASA open-to-close ratios for the S2H14, CR3022, S2H13 epitopes and NIDS shown in green (>1) and purple (<1). Diagrams show ASA ratios for single epitope residues. Average ASA-ratio change over all epitope residues is given in %. **f** Retention assay for MiniVs presenting native S and incubated with MCF7 cells for 24 h. Reduction in retention by ADAH11 nanobodies was measured by addition of 1 μM ADAH11 during the incubation period. **g** Native S-normalized neutralization of ADAH11 for MiniVs presenting FFA-loaded ApoS incubated with MCF7 cells for 24 h. **h** Retention assay for MiniVs presenting native S and incubated with MCF7 cells for 24 h. Reduction in retention by CR3022 IgG was measured by addition of 132 nM CR3022 during the incubation period. **i** Native S-normalized neutralization of CR3022 based for MiniVs presenting FFA-loaded ApoS incubated with MCF7 cells for 24 h. **j** Retention analysis with native S-normalized neutralization of MiniVs by convalescent COVID-19 patient serum-derived IgGs under low (0.1 μM) and high (1 μM) LA/AA concentrations. **k** Model of FFA as molecular switches that couple local inflammatory states to SARS-CoV-2 S immunogenicity. Results in **f–j** are shown as mean ± SD from at least $n = 3$ biological replicates in each experimental condition, *$p < 0.05$, **$p < 0.005$, ***$p < 0.0005$, unpaired two-tailed $t$-test. Source data are provided as a Source Data file.

inflammatory phases. This is achieved through coupling the RBD open-to-close equilibrium to the abundance of FFAs. In this context, the RBD´s major contribution to SARS-CoV-2 immunogenicity and its fundamental importance for COVID-19 vaccination strategies has been recognized by several studies[5,6,41,48]. Fatty acids are exceptionally suitable metabolic markers for immunogenic-host adaptation as altered fatty acid metabolism is an early indicator for beginning antiviral immune response against positively stranded RNA-viruses[49]. This enables efficient viral replication and transmission during the initial incubation period until progressive activation of the host´s antiviral response. After immune recognition, FFA-binding could mediate immune-evasive adaptation, suppress SARS-CoV-2 immunogenicity and reduce viral infection via a stealth-like mechanism of reduced RBD exposure[40]. Eventually, this could lead to increased viral titers because of reduced virus clearance. In summary, enabled by our MiniV technology, we identified a potentially immune-evasive link between FABP and FFAs in SARS-CoV-2 S which could be exploited for future COVID-19 therapy.

## Methods

**Materials**. 18:1 DOPC 1,2-dioleoyl-sn-glycero-3-phosphocholesteroline, 18:1 DOPE 1,2-dioleoyl-sn-glycero-3-phosphoethanolamine, L-α-phosphatidylglycerol (EggPG), L-α-phosphatidylcholine (EggPC), LissRhod PE 1,2-dioleoyl-sn-glycero-3-phosphoethanolamine-N-(lissamine rhodamine B sulfonyl), 18:1 DGS-NTA(Ni) 1,2-dioleoyl-sn-glycero-3-[(N-(5-amino-1-carboxypentyl)iminodiacetic acid)succinyl] (nickel salt), 18:1 1,2-dioleoyl-sn-glycero-3-phospho-(1'-myo-inositol) (ammonium salt), 18:1 1,2-di-(9Z-octadecenoyl)-sn-glycero-3-phospho-L-serine (sodium salt), 1,2-Dioleoyl-sn-glycero-3-phosphoethanolamin Atto488-conjugate, 18:1, cholesterol, 18:0 N-stearoyl-D-erythro-sphingosylphosphorylcholine, and extrude set with 50 nm pore size polycarbonate filter membranes were purchased from Avanti Polar Lipids, USA Dulbecco's Modified Eagle Medium (DMEM) High Glucose, heat-inactivated fetal bovine serum, penicillin-streptomycin (10,000 U/mL), L-Glutamine (200 mM), trypsin-EDTA (0.05%), FluoroBrite DMEM, Cell-Tracker Green CMFDA dye, Hoechst 33342, phosphate-buffered saline were purchased from Thermo Fischer Scientific, Germany. Dexamethasone (BioReagent), Tretinoin (pharmaceutical standard), Acitretin (>98%), Vitamine K₂, Carbenoxolone disodium salt (>98%), palmitic acid (>99%), oleic acid (>99%), heparin, linoleic acid (>99%) arachidonic acid (analytical standard), oxalyl chloride (>99%), 3-picolylamine (>99%) were purchased from Sigma Aldrich, Germany. LinRGD was custom synthesized by PSL GmbH, Germany. Transparent flat-bottom 96-well plates were purchased from TTP, Switzerland. Human Insulin was purchased from Millipore Sigma. Lipidex-1000 resin was purchased from Perkin Elmer, Germany. DMSO for cell culture use was purchased from Omnilab, Germany. Sicastar Ni²⁺-NTA silica beads were purchased from Micromod GmbH, Germany. Human IgG COVID-19 convalescent plasma fractionated purified and lyophilized was purchased from Innovative Research, USA. Anti SARS-CoV-2 S, clone CR3022 (FITC) was purchased from Novus Biologiclas, Germany. Histidine-tagged recombinant SARS-CoV-2 spike RBD (Val16-Arg685(D614G)), human recombinant ACE2 were purchased from Sino Biologicals, Germany. MCF-7 cells, HUVEC cells and endothelial cell growth medium were obtained from ATCC, USA. QCM-D sensor crystals (QS-QSX303) were obtained from Quantum Design GmbH, Germany.

**SUV/MiniV preparation**. SUVs and MiniVs were produced by manual extrusion through track-etched polycarbonate filter membranes[50,51]. For this, lipids dissolved in chloroform stock solutions were mixed at the desired lipid ratio in glass vials and subsequently dried under vacuum for at least 15 min to evaporate the chloroform completely. The obtained lipid film was rehydrated to a final lipid concentration of 6 mM in PBS for at least 5 min and afterwards shaken for at least 5 min at 1000 rpm on a horizontal shaker. This liposome solution was extruded nine times through a 50 nm radius pore size filter.

For immobilization of recombinant S ectodomains, the SUV solution was diluted to a final concentration of 100 μM in PBS, corresponding to 1 μM final DGS-NTA(Ni²⁺). To this, 0.5 μM of recombinant histidine-tagged S was added and incubated for coupling for at least 15 min. For loading of FAs on MiniVs, FAs were dissolved in DMSO to a final concentration of 100 mg/mL. From these stocks, 100 μg/mL dilutions in PBS were prepared.

**S proteins expression and purification**. Recombinant wild-type (Wuhan) S protein with a mutated furin site was produced as described previously[1]. The construct comprises amino acids 1–1213, lacks the native transmembrane domain, which is replaced with a C-terminal thrombin cleavage site followed by a T4-foldon trimerization domain and a hexa-histidine tag. The polybasic cleavage site has been removed (RRAR to A mutation). Briefly, S protein was expressed using the

MultiBac baculovirus expression system in Hi5 cells[52]. Medium from transfected cells was harvested 3 days after-transfection by centrifuging the cultures at $1000 \times g$ for 10 min followed by another centrifugation of supernatant media at $5000 \times g$ for 30 min. This final supernatant was then incubated with 10 mL HisPur Ni-NTA Superflow Agarose (Thermo Fisher Scientific) per 4 l of expression culture for 1 h at 4 °C. Subsequently, the resin-bound with SARS-CoV-2 S protein was collected using a gravity flow column (Bio-Rad) and then extensively washed with 30 column volumes (CV) of wash buffer (65 mM NaH₂PO₄, 300 mM NaCl, 20 mM imidazole, pH 7.5). Finally, the protein was eluted using a step gradient of elution buffer (65 mM NaH₂PO₄, 300 mM NaCl, 235 mM imidazole, pH 7.5). After analysing elution fractions using reducing SDS-PAGE, fractions containing SARS-CoV-2 S protein were pooled and concentrated using 50 kDa MWCO Amicon centrifugal filter units (EMD Millipore) and finally buffer-exchanged in phosphate-buffered saline (PBS) pH 7.5. The protein was then subjected to size exclusion chromatography (SEC) using a Superdex 200 increase 10/300 column (GE Healthcare) in PBS pH 7.5. Peak fractions from SEC were analysed using reducing SDS-PAGE and then fractions containing SARS-CoV-2 S protein were pooled and concentrated using 50 kDa MWCO Amicon centrifugal filter units (EMD Millipore) and finally aliquoted and flash-frozen in liquid nitrogen for storage at −80 °C until further use.

The UK (or 'Kent') B1.1.7 variant[53] S ectodomain gene sequence was synthesized and inserted into pACEBac1 plasmid (Genscript Inc., New Jersey USA). Expression and purification were carried out as described above for wild-type S.

The S(R403A) mutant was prepared by modifying the wild-type (Wuhan) S expression construct with the point mutation using the QuickChange site-directed mutagenesis kit (Qiagen).

SARS-CoV S encoding gene was synthesized (Genscript Inc, New Jersey USA). The construct comprises amino acids 14 to 1193, preceded by a GP64 secretion signal sequence (amino acids MVSAIVLYVLLAAAAHSAFA) and contains a C-terminal thrombin cleavage site followed by a T4-foldon trimerization domain and a hexa-histidine affinity purification tag. The synthetic gene was inserted into pACEBac1s[52]. Protein was produced and purified as described above for SARS-CoV-2 S.

MERS-CoV S encoding gene was synthesized (Genscript Inc, New Jersey USA) and cloned into pACEBac1[52]. This construct comprises amino acids 18–1294, preceded by the GP64 secretion signal sequence and contains a C-terminal thrombin cleavage site followed by a T4-foldon trimerization domain and a hexa-histidine affinity purification tag. Protein was produced and purified as described above for SARS-CoV-2 S.

ApoS protein lacking free fatty acid was produced from purified SARS-CoV-2 S by Lipidex treatment as described[1]. Briefly, purified SARS-CoV-2 S protein was incubated with pre-equilibrated lipidex-1000 resin (Perkin Elmer; cat no. 6008301) in PBS pH 7.5 overnight at 4 °C on a roller shaker. Following this, Lipidex-treated S protein was separated from the resin using a gravity flow column. The integrity of the protein was confirmed by size exclusion chromatography (SEC) using a S200 10/300 increase column (GE Healthcare) and SDS-PAGE.

**ADAH11 selection, expression, and purification**. Neutralizing nanobody ADAH11 against the RBM of S was selected from a synthetic library using in vitro selection by ribosome display[54]. Following selection, the ADAH11 coding sequence was cloned into pHEN6 plasmid[55] containing a PelB signal sequence at the N-terminus, and a hexa-histidine and 3X FLAG tag at the C-terminus. ADAH11 was expressed in E. coli TG1 cells in 2x YT medium overnight at 30 °C induced with 1 mM IPTG (Isopropyl ß-D-1-thiogalactopyranoside). Cells were harvested by centrifugation at $3200 \times g$ for 15 min at 4 °C. Cells were then resuspended in 5 mL cold TES buffer (50 mM Tris pH 8.0, 20% Sucrose, 1 mM EDTA, complete protease inhibitor tablet) for each gram of pellet. This resuspension was incubated on a roller shaker for 45 min at 4 °C. Then, 7.5 mL of ice-cold shock buffer (20 mM Tris pH 8.0, 5 mM MgCl₂) was added per gram of pellet and again incubated on a roller shaker for 45 min at 4 °C. Then, the supernatant-containing periplasm was collected by centrifugation at $13,000 \times g$ for 12 min at 4 °C. This supernatant was incubated with 0.5 ml HisPur Ni-NTA Superflow Agarose (Thermo Fisher Scientific) per liter of expression for 1 h. The resin was pre-equilibrated in ADAH11 wash buffer 1 (50 mM Tris, 200 mM NaCl, 10 mM Imidazole pH 8.0). A gravity flow column (Bio-Rad) was used to separate the resin-bound with ADAH11 from the unbound lysate. This resin was then washed with 20 CV of ADAH11 wash buffer 1, followed by 30 CV wash with ADAH11 wash buffer 2 (50 mM Tris pH 8.0, 200 mM NaCl, 20 mM Imidazole pH 8.0). Protein was then eluted using a step gradient of ADAH11 elution buffer (50 mM Tris, 200 mM NaCl, 500 mM Imidazole pH 8.0). Elution fractions were analyzed using reducing SDS-PAGE. Elution fractions containing ADAH11 protein were pooled and dialyzed against PBS pH 7.5. Dialyzed protein was concentrated using 10 kDa MWCO Amicon centrifugal filter units (EMD Millipore) and then injected on a Superdex 200 increase 10/300 size exclusion chromatography (SEC) column (GE Healthcare) in PBS pH 7.5. Peak fractions from SEC were analyzed using reducing SDS-PAGE and fractions containing ADAH11 were pooled and concentrated using 10 kDa MWCO Amicon centrifugal filter units (EMD Millipore). Finally, the protein was aliquoted and flash-frozen in liquid nitrogen for storage at −80 °C until further use.

**DLS + zeta potential**. Size and zeta potentials of SUVs and MiniV variants were measured with a Malvern Zetasizer Nano ZS system at a total lipid concentration of 100 μM in PBS[56]. Temperature equilibration time was set to 300 s at 25 °C, followed by three repeat measurements for each sample at a scattering angle of 173° using the built-in automatic run-number selection. The material refractive index was set to 1.4233 and solvent properties to $\eta = 0.8882$, $n = 1.33$ and $\varepsilon = 79.0$.

**Hoechst staining and nuclei counting**. To assess blocking of cell adhesion under linRGD incubation, we imaged adherent cells after seeding and washing on fibronectin-coated well plates following previously developed protocols[57,58]. To this end, MCF-7 cells were seeded at a density of 50,000 cells/well in flat-bottom 96-well plates in 100 μL culture medium. Cells were either incubated with 20 μM linRGD or with the addition of 20 μL PBS (mock) for 24 h. Subsequently, Hoechst33342 was added to a final concentration of 10 μM to the cell layers and incubated for 10 min. Cells were then washed twice with 200 μL PBS and fixed for 20 min with 4% paraformaldehyde. Cell nuclei were then imaged in the whole well with a Leica DMi8 inverted fluorescent microscope equipped with a sCMOS camera and 10× HC PL Fluotar (NA 0.32, PH1) objective with DAPI emission/excitation filters. For automated nuclei counting, TIFF images from three wells (i.e., three replicates) were background segmented by global histogram thresholding and automated particle counting (particle analysis) with ImageJ software (NIH). Before particle counting, a watershed algorithm was applied to separated overlapping nuclei and nuclei counting was restricted to particles in the size range between 1 μm² and 100 μm².

**Cryo-TEM tomography**. For cryo-TEM imaging, MiniVs were diluted to a final particle concentration of $2 \times 10^{10}$ particles/mL. The samples were applied to glow-discharged C-Flat 1.2/1.3 4 C grids (Protochips) and plunge-frozen using a Vitrobot Mark IV (FEI, now Thermo Scientific) at the following settings: temperature of 22 °C, the humidity of 100%, 0 s wait time, 2–3 s blot time, blot force −1, and 0 s drain time. The data were acquired on a Titan Krios electron microscope operated at 300 kV equipped with a Falcon 3EC direct detector, a Volta phase plate, and a Cs Corrector (CEOS GmbH). The micrographs were acquired with the software EPU (Thermo Scientific) in linear and counting mode at magnified pixel sizes of 1.39 Å and 0.85 Å, at doses of 40–60 e-/sqÅ, at defocus values ranging from −0.25 to −2.0 μm using the Volta phase plate or the 100 μm objective aperture. Tilt series were acquired with the software TOMO (Thermo Scientific) at tilt angles ranging from −60° to 60° at 3° increments with defocus values ranging from −0.3 to −1.0 μm with individual micrograph movies recorded with 10 frames in counting mode at a magnified pixel size of 0.85 Å, a total dose per micrograph of 3e-/sqÅ, using the Volta phase plate. Micrograph movies were aligned using MotionCor2 in the RELION3 software suite[59,60]. Tilt series were binned by a factor of four for further processing in Etomo (IMOD)[61]. For each tilt series, the micrographs were aligned using patch tracking. The final tomograms were generated with a filtered back-projection of the aligned micrographs. Tomogram slices were FFT band-pass filtered between 3 and 40 pixel and subsequently a $2 \times 2$ pixel Gaussian blur filter was applied. Images were contrast corrected by visual inspection.

**QCM-D + imaging SLB**. For QCM-D measurements, sensor crystals, AT-cut gold electrodes coated with a 50 nm thick layer of silicon oxide were used. SiO₂ surfaces were cleaned as described elsewhere[62]. Briefly, QCM-D sensor crystals were cleaned using an aqueous 2% SDS solution water and activated by UV/ozone for 10 min. A QSense Analyzer equipped with a four-channel system from QuantumDesign was used for measurements. All measurements were performed at 22 °C in an open mode. The resonance frequency and dissipation shifts were recorded at several harmonics simultaneously. Before adding the samples, the frequency and dissipation changes were base-lined by averaging over the last 5 min of the buffer wash (PBS). Formation of supported lipid bilayers (SLBs) was achieved by absorption and rupture of SUVs on cleaned SiO₂ surfaces. SUVs with a lipid composition of 20 mol% L-α-phosphatidylglycerol (EggPG), 77 mol% L-α-phosphatidylcholine (EggPC), 2 mol% 1,2-Dioleoyl-sn-glycero-3-phosphoethanolamin, 2 mol% 1,2-dioleoyl-sn-glycero-3-[(N-(5-amino-1-carboxypentyl)iminodiacetic acid)succinyl] (18:1 DGS-NTA), and 1 mol% 1,2-Dioleoyl-sn-glycero-3-phosphoethanolamin Atto488-conjugate were used for SLB formation at a lipid concentration of 1.2 mM and a final MgCl₂ concentration of 2 mM. Two hundred microliters of this solution was added to each QCM-D sensor to form SLBs after the sensors were equilibrated with 200 μL PBS for 7 min and incubated for 17 min. SLB formation was confirmed by the characteristic changes in frequency and dissipation as it is described elsewhere[63]. SLBs were washed with PBS to remove non-ruptured SUVs for 7 min. For immobilization of recombinant human AC2 200 μL PBS containing 300 nM histidine-tagged ACE2 was added to the according sensor crystals (S1, S2, and S4) and incubated for 16 min. Unbounded ACE2 receptors were removed washing with PBS. The crystal sensor S3 not being functionalized with the ACE2-receptor was treated in the same way with the washing buffer (5). After 11 min, the washing buffer was removed and 200 μL of a solution containing 1.5 μM MiniVs (total lipid concentration) was added to sensors S1 and S3. To the crystal of sensor S2, a 200 μL PBS solution containing 1.5 μM (total lipid concentration) naive SUVs was added. Sensor S4 was treated with

200 μL PBS containing soluble recombinant S. The solutions were incubated on the crystal sensors for 55 min and afterwards washed with PBS to remove unbound components (7).

**Cell culture and viruses**. MCF-7 cells were cultured in Dulbecco's Modified Eagle Medium supplemented with 4.5 g/l glucose, 1% L-glutamine, 1% penicillin/streptomycin, 0.01 mg/mL recombinant human insulin, and 10% fetal bovine serum. HUVEC cells were cultured in F-12K medium supplemented with 0.1 mg/ml heparin, 10% fetal bovine serum and 30 μg/ml endothelial growth supplement. Cells were routinely cultured at 37 °C and 5% CO₂ atmosphere and passaged at ~80% confluency by detachment with 0.05% trypsin/EDTA treatment. MCF-7 cells were cultured in Dulbecco's Modified Eagle Medium (DMEM) supplemented with 4.5 g/l glucose, 1% L-glutamine, 1% penicillin/streptomycin, 0.01 mg/mL recombinant human insulin, and 10% fetal bovine serum. A549-ACE2 cells[24] were kindly provided by M. Cortese and R. Bartenschlager (Heidelberg University) and cultivated in DMEM supplemented with 10% fetal calf serum (Capricorn), 100 U/mL penicillin, 100 μg/mL streptomycin, and 1% non-essential amino acids (all from Gibco) and 0.5 mg/mL G418. Cells were routinely cultured at 37 °C and 5% CO₂ atmosphere and passaged at ~80% confluency based on 0.05% trypsin/EDTA treatment.

The SARS-CoV-2 isolate Bavpat1/2020 was obtained by the European Virology Archive (Ref-SKU: 026V-03883) at passage 2. Virus stocks were generated by passaging the virus two times in VeroE6 cells. Virus stocks were titrated by infection of VeroE6 cells as previously described[24].

**Confocal microscopy**. Confocal microscopy was performed with a laser-scanning microscope LSM 800 (Carl Zeiss AG). Images were acquired with a 20x (Objective Plan-Apochromat ×20/0.8 M27, Carl Zeiss AG) and a ×63 immersion oil objective (Plan-Apochromat ×63/1.40 Oil DIC, Carl Zeiss AG). Images were analyzed with ImageJ (NIH) and adjustments to image brightness and contrast, as well as background corrections, were always performed on the whole image and special care was taken not to obscure or eliminate any information from the original image.

**Mass spectrometry**. For loading of ApoS samples with defined FFA for MS analysis, pure FFA (PA, OA, LA, AA) solutions were prepared at a stock concentration of 100 mg/mL in DMSO. Then, a predilution of 100 μg/mL in PBS was prepared and 0.2 mg ApoS samples were mixed with respective FFA predilutions in a 1:10 molar ratio and incubated for 2 h. Subsequently, the histidine-tagged ApoS protein was pulled down with 1 mg of 300 nm NTA(Ni2 + )-conjugated silica beads through 30 min incubation and subsequent centrifugation at $11,000 \times g$ for 5 min. ApoS was eluted by the addition of 100 mM imidazole and subsequently subjected to MS samples preparation (see below). For mass spectrometry detection of the FA content in S samples, a targeted LC-coupled MS/MS approach with multiple reaction monitoring was applied. For MS analysis, all protein samples were pre-diluted to a final concentration of 400 μg/mL. To extract the FAs from the S protein, S samples were mixed in a 1:4 (v/v) ratio with chloroform for 2 h on a horizontal shaker in a Teflon-sealed glass vial at 25 °C. The top organic phase was then transferred to a new glass vial and the chloroform was evaporated for 15 min in a desiccator. Subsequently, a derivatization approach enabling improved MS detection of FFAs was applied(2). This approach is based on the activation of carboxylic FA head groups with oxalyl chloride (OC) and subsequent derivatization with 3-picolylamine (PA). For this, 200 μL of a 2 M OC solution prepared in dichloromethane (DCM) was added to the samples using a glass syringe and incubated for 5 min at 65 °C. The OC and DCM were then removed in a desiccator and 150 μL of a 3-picolylamine solution (1% (v/v) in acetonitrile) was added to the samples and incubated for 5 min. The residual solvent was again evaporate in a desiccator and the samples were dissolved in 50 μL acetonitrile. To all samples, a deuterated LA internal standard (dissolved in DCM) was added to a final concentration of 20 μg/mL before derivatization. Finally, the resulting samples were diluted 1–500 in 80% ACN aq. (MS grade).

The subsequent LC-MSMS analysis was performed using a Shimadzu Nexera system hyphenated to a Sciex QTRAP 4500 system. The LC system was supplied with water (A) and acetonitrile (B) in LCMS grade from Biosolve. The solvents were supplemented with 0.1% LCMS grade formic acid. 6 μL of the sample solutions were fractionated with a Supelco Titan C18 column 100 × 2.1 mm, 1.9 μ, at 45 °C. Whereas the following gradient was applied: 0 min—40%B; 1.5 min: 40% B; 6.5 min—98%B; 8.0 min—98%B; 8.1 min—40%B and 9.5 min—98% B. The mass spectrometer was controlled using the Analyst 1.7 software and was operated in ESI positive mode. An optimal target compound ionization was achieved by setting the following source parameters: Curtain gas 35, temperature 550 °C, ionization voltage 5500 V, nebulizer gas 65, heater gas 80 and collision gas 9.

The targeted derivatized fatty acids were detected in MRM mode. Fragmentation of the monitored PA derivatized FA yielded a characteristic MSMS fragment with m/z 92. For a linoleic acid additional parent to fragment MSMS transitions were recorded. The following compound dependent MSMS parameters were applied: PA derivatized linoleic acid (LAP-92: 371.3–92.1 Da, Dwell 90 ms, declustering potential (DP) 80, entrance potential (EP) 10, collision energy (CE) 43, cell exit potential (CXP) 11, LAP-109: 371.3–109.1 Da, Dwell 40 ms, DP 80, EP 10,

CE 36, CXP 10, LAP-163: 371.3–163.1 Da, Dwell 15 ms, DP 80, EP 10, CE45, CXP 11); PA derivatized deuterated linoleic acid (LADP-92: 375.3–92.1 Da, Dwell 50 ms, DP 80, EP 10, CE 43, CXP 11); PA derivatized palmitic acid (PAP-92: 347.3–92.1 Da, Dwell 50 ms, DP 80, EP 10, CE 43, CXP 11); PA derivatized oleic acid (OAP-92: 373.3–92.1 Da, Dwell 50 ms, DP 80, EP 10, CE 43, CXP 11); PA derivatized arachidonic acid (AAP-92: 395.3–92.1 Da, Dwell 50 ms, DP 80, EP 10, CE 43, CXP 11). Subsequent data analysis was performed using the Analyst and Multiquant 3.0.2 software.

A set of positive and negative control samples was measured under the same conditions to ensure high-quality results. As positive controls derivatized LA standard dilutions in 80% ACN aq. (MS grade) starting from the following concentration levels were used: 5, 10, and 25 μg/mL. Four-fold injection of the 25 μg/mL LAP standard yielded a CV < 2%. A sample preparation related contamination of the samples was excluded using a negative control sample, that was generated by using a water aliquot instead of the protein solution for the sample work-up. Moreover, we ensured the chromatographic separation of LADP (C18H28D4O2) and the ubiquitous as well as isobaric stearic acid (C18H36O2).

**Retention assays.** For quantification of MiniV-cell-binding, we developed retention assays to measure the amount of MiniVs retained within culture plates after incubation and washing. This assay is based on quantification of the rhodamine fluorescence form the SUV lipid membrane and can therefore not discriminate between attachment and uptake of MiniVs. For retention analysis, SUVs and MiniVs with different recombinant S ectodomains (as indicated in the figure legends), were added to MFC-7 cell cultures in flat-bottom 96-well plates with 100 μL culture medium to a final lipid concentration of 10 μM. Before the addition, cell medium was exchanged from the seeding medium (DMEM supplemented with phenol red, 4.5 g/l glucose, 1% L-glutamine, 1% penicillin/streptomycin, 0.01 mg/mL recombinant human insulin, and 10% fetal bovine serum) to low-serum medium (DMEM supplemented without phenol red, 4.5 g/l glucose, 1% L-gluta-mine, 1% penicillin/streptomycin, 0.01 mg/mL recombinant human insulin, and 0.5% fetal bovine serum) to reduce the amount of serum-derived FAs far below physiologically relevant concentrations. After incubation of MiniVs with cells for 24 h, rhodamine fluorescence was measured at 9 different positions and 1300 μm distance to well wall in each well using an Infinite M200 TECAN plate reader controlled by TECAN iControl software with an in-built gain optimization and excitation/emission setting adjusted to 555/585 nm. Wells were then washed three times with PBS and subsequently fixed with 100 μL 4% paraformaldehyde. After 10 min fixation, rhodamine fluorescence was again measured in each well with the settings mentioned above. For retention analysis binding could then be deduced from residual fluorescence in each well (for assessment of patient IgG neutralization) or retention values could be calculated by dividing the residuals fluorescence to the initial fluorescence intensity before washing. All measurements were performed in triplicates. As retention show significant variations depending on cell seeding density, cell viability and vesicle preparation, a condition of naïve SUVS (i.e., without protein on the surface) was added to all experimental batches for normalization proposes. For time-resolved measurements of retention, separate wells for each time point were prepared and evaluated sequentially.

For retention assay involving an assessment of FFA influence, stock of pure FFA (PA, OA, LA and AA) were prepared in DMSO at a final concentration of 100 mg/mL. From these stock, predilutions (100 μg/mL) in PBS were prepared. Individual FFAs, as indicated in the figures, were added to the MiniVs of the retention assays at a final concentration of 1 μM. For retention assays involving TMPRSS2 inhibition by camostat mesylate, camostat mesylate was added to the cell cultures at indicated concentration 2 h prior to MiniV addition.

**Competition experiments of D614G and B1.1.7S.** For assessment of competitive cell-binding between MiniVs presenting WT(D614G) and B1.1.7 S on the surface, two differently fluorescent SUV samples were prepared. SUVs harbouring Atto448 we decorated with WT(D614G) S and SUVs harbouring rhodamine B with B1.1.7 S. 10 μM final lipid concentration of both MiniV types were incubated together with MCF-7 cells and retention assays were performed as described above for the rhodamine and Atto488 (emission excitation set to 488/512) signals separately. Naïve SUV controls were performed for both vesicles types and retention were normalized accordingly.

Based on the differing fluorescence of the two MiniV populations, multiplicity of infection (MOI) measurements were performed by confocal imaging. For this, both MiniV types (D614G and B1.1.7) were incubated with MCF-7 cells in glass-bottom 8-well LabTek chambers for 24 h. After incubations, cells were washed three times with PBS and subsequently imaging in at least 8 z-positions (1 μm stack distance) in both channels. Maximum intensity projects were made by ImageJ and MiniVs of each type were manually counted for 5 individual cell groups.

**Retention assays (RGD).** For assessment of RGD-motif-mediated effects in MiniV-cell-binding, retention assay under integrin blocking conditions with linRGD was performed. For this, 100,000 MCF-7 cells/well were seeded in flat-bottom 96-well plates and allowed to form confluent monolayers overnight. Subsequently, the seeding medium was exchanged to low-serum (0.5%) cell culture medium and a final linRGD concentration of 20 μM was added to each well from a

10 mM stock solution. MiniVs (with S-configurations as indicated in the figure legends) were added to a final lipid concentration of 10 μM. Retention assays were performed as detailed above.

To quantify differences in FA-regulated S-integrin engagement, retention was calculated as linRGD-normalized retention. For this, retention of MiniVs was measured with and without incubation of linRGD. Differences in retention were calculated and normalized by:

$$\frac{R - R_{\text{linRGD}}}{R_{\text{nativeS}}}$$

Where R is the retention value of the respective MiniV S configuration without linRGD, $R_{\text{linRGD}}$ is the retention value of the corresponding MiniV S configuration under the addition of 20 μM linRGD and $R_{\text{nativeS}}$ is the retention value MiniVs presenting native S.

**Retention assays (FABP drug assessment).** For assessment of drug-modulated S binding of potential pharmacologic FABP binders, retention assays under drug incubation were performed. For this, 100,000 MCF-7 cells/ well were seeded in flat-bottom 96-well plates and allowed to form confluent monolayers overnight. Subsequently, the seeding medium was exchanged to low-serum (0.5%) cell culture medium and a final drug concentration of 1 μM was added to each well from DMSO stock solutions. MiniVs (with S-configurations as indicated in the figure legends) were added to a final lipid concentration of 10 μM and incubated for 24 h. For normalization purposes, also retention of naïve SUVs under drug treatment was measured to account for any drug-induced changes in cellular phenotypes. Retention assays were performed as detailed above and drug-normalized retention was calculated from:

$$\frac{R * R_{\text{corr}}}{R_{\text{apo}}} \quad (1)$$

where $R_{\text{corr}}$ is calculated by

$$\frac{R_{\text{SUVs}}}{R_{\text{drugSUVs}}} \quad (2)$$

and R is is the retention value of the respective MiniV ApoS configuration under drug incubation, $R_{\text{apo}}$ is the retention value of MiniVs with ApoS without drug incubation, $R_{\text{SUVs}}$ is the retention value of SUVs without drug incubation and $R_{\text{drugSUVs}}$ is the retention value of SUVs under drug incubation. IC50 values of MiniV retention for dexamethasone and vitamin K2 were measured by serial dilutions and IC50 values were calculated using nonlinear regression.

**Protein structure visualization and ASA calculation.** Previous studies have demonstrated the value of accessible surface area calculations for assessment of IgG epitope characterization in SARS-CoV-2 S and conformational changes upon protein binding[64,65]. For visualization of S cryo-TEM structures, the PDB 3D viewer was applied[66] and protein structure was retrieved from https://www.wwpdb.org/pdb?id=pdb_00007bnn; https://www.wwpdb.org/pdb?id=pdb_00006zb5; https://www.wwpdb.org/pdb?id=pdb_00007a97. ASAs were calculated from molecular surface representations of residue accessible surface area properties in the PBD. A rolling probe radius of 1.4 nm was applied. For ASA calculations hydrogen atoms were taken into consideration. Tracing atoms were not considered for the analysis and no line size attenuation was applied.

**Nanoparticle tracking analysis.** MiniV particle concentration was determined by NTA with a ZetaView Quatt Video Microscope PMX-420 (Particle Metrix, Inning am Ammersee, Germany). Alignment was performed using 100 nm polystyrene beads diluted 1:250 000 (v:v) in MilliQ water. MiniV particles were diluted in PBS to a final concentration of 50–150 particles/frame. The observation cell was equilibrated with PBS before 1 mL of the sample was injected. One acquisition cycle was performed at 11 positions of the observation cell, in scatter mode using the 488 nm laser and at a temperature of 24 °C. The following settings were used for acquisition: sensitivity 80, shutter 100, frame rate 30, medium quality. Post-acquisition parameters were set as follows: minimal brightness 30, minimal area 10, maximal area 1000, trace length 15. All 11 positions were included in the analysis, using over 500 traced particles in total.

**MiniV neutralization assays.** For MiniV neutralization assays, 100,000 MCF-7 cells/ well were seeded in flat-bottom 96-well plates and allowed to form confluent monolayers overnight. Subsequently, the seeding medium was exchanged to low-serum (0.5%) cell culture medium and MiniVs (with S-configurations as detailed in the figure legends) were added to a final lipid concentration of 10 μM. Three types of neutralizing immunoglobulins were tested (1) IgG CR3022 (2) ADAH11, and (3) purified IgG from convalescent COVID-19 donors. CR3022 neutralization was assessed by the addition of 132 nM purified IgG. ADAH11 neutralization was titrated in a concentration range of 7.4 nM–1.5 μM and retention assays for ADAH11 neutralization were performed at 1 μM ADAH11. Neutralization of purified donor IgGs was measured at a final concentration of 3.3 μg/mL.

For assessment of neutralization with different FFA profiles, immunoglobulin mediated reduction in retention was calculated by:

$$\frac{R - R_{immunoglobulin}}{R_{NativeS}} \qquad (3)$$

where R is the retention value of the respective MiniV ApoS configuration without immunoglobulin, $R_{immunoglobulin}$ is the retention value of the respective MiniV ApoS configuration under immunoglobulin incubation and $R_{NativeS}$ is the retention value of MiniVs presenting Native S without immunoglobulin incubation. For the assessment of MiniV neutralization as a function of FFAs, individual FFAs were added from DMSO stocks (100 mg/mL) to the culture medium of the retention assay at a final concentration of 1 μM. To mimic the basal, low FFA levels, for COVID-19 donor IgG neutralization, no additional FFAs were added to the culture medium as the 0.5% serum concentration of the retention assay culture medium already provide approximately 0.1 μM LA and AA[67].

**Competition assays.** $5 \times 10^4$ A549-ACE2 cells were seeded in 24-well plates on the day prior to infection. Cells were incubated with 300 μL of SUV or MiniV dilution for 2.5 h. 100 μL of virus suspension containing $5 \times 10^4$ infectious virus particles (multiplicity of infection of 1) were added to each well and incubated for 2 h. The medium was removed and cells were washed two times with sterile PBS before the addition of 1 mL of fresh medium. Cells were harvested 18 h post-infection and total RNA extracted using NucleoSpin RNA Plus kit (Macherey-Nagel), following manufacturer's instructions. cDNA was generated using the High-Capacity cDNA Reverse Transcription kit (Applied Biosystems) following the manufacturer's instructions. Expression levels of GAPDH and SARS-CoV-2 Orf7a mRNA were determined by using the iTaq Universal SYBR Green 2x (Bio-Rad). Reactions were performed on an CFX96 (Bio-Rad) using the following program: 95 °C for 3 min and 45 cycles as follows: 95 °C for 10 s, 60 °C for 30 s. GAPDH mRNA level was used for the normalization of input RNA. Relative abundance of each specific mRNA was determined by using the ΔΔCT method as previously described[68]. The following primers were used:

GAPDH-For 5′ - GAAGGTGAAGGTCGGAGTC - 3′
GAPDH-Rev 5′ - GAAGATGGTGATGGGATTTC - 3′
CoV-2 Leader_For 5′ - TCCCAGGTAACAAACCAACCAACT- 3′
CoV-2 Orf7a_Rev 5′ - AAATGGTGAATTGCCCTCGT- 3′.

**Live virus neutralization assays.** Purified IgG antibodies were serially diluted 2-fold in Opti-MEM, starting with a dilution of 1:20 and mixed with an equal volume of Opt-MEM containing $5 \times 10^4$ pfu SARS-CoV-2 (final multiplicity of infection of 1). IgGs/virus mixes were incubated for 1 h at 37 °C and subsequently transferred to 24-wells containing $5 \times 10^4$ A549-ACE2 cells seeded the day prior to infection. Cells were infected for 2 h at 37 °C, subsequently washed once with sterile PBS, and cultured for a further 6 h in a fresh medium. Cells were washed and harvested for RNA extraction and ORF7a mRNA expression levels were quantified by qRT-PCR as described above. Values were normalized to infection levels in absence of IgG antibodies. Relative inhibitory concentration of 50% (IC50) values were calculated using nonlinear regression.

**Reporting summary.** Further information on research design is available in the Nature Research Reporting Summary linked to this article.

## Data availability

The data generated in this study are available in the main text, the supplementary materials, the source data file or the corresponding authors upon reasonable request. Source data are provided as separate source data file. Protein structures used in this study were retrieved from the protein data bank under the accession codes 7BNN, 6ZB5, and 7A97. Source data are provided with this paper.

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

## Acknowledgements

We would like to thank Sabine Grünewald for cell culture support, Isabelle Kajzar for critical discussion of the data, Ulrike Mersdorf for support in MiniV negative staining and TEM imaging and Mirko Cortese and Christopher J. Neufeldt (Molecular Virology Heidelberg) for sharing their expertise with SARS-CoV-2 infection. Some elements in the figures were created with BioRender.com. Support from the Heidelberg Bioscience International Graduate School and the Max Planck School Matter to Life is acknowledged by O.S. J.P.S. is the Weston Visiting Professor at the Weizmann Institute of Science and part of the excellence cluster CellNetworks at the University of Heidelberg. O.S. is the Meurer Visiting Professor at the University of Bristol. The Max Planck Society is appreciated for its general support by O.S., J.E.H.B., A.Y.R., M.M., S.F., E.A.C.A., I.P., J.P.S. J.P.S. and I.P. acknowledge funding from the Federal Ministry of Education and Research of Germany, Grant Agreement no. 13XP5073A, PolyAntiBak and the MaxSynBio Consortium, which is jointly funded by the Federal Ministry of Education and Research of Germany and the Max Planck Society. J.P.S. and I.P. also acknowledge the support from the Volkswagen Stiftung (priority call 'Life?'). The German Science Foundation SFB1129 (project nr. 240245600-SFB1129 P15 to A.E.C.A and J.P.S. and P13 to J.P.S.) is acknowledged by E.A.C.A., A.R., and J.P.S. J.P.S acknowledges the support from Germany's Excellence Strategy via the Excellence Cluster 3D Matter Made to Order (EXC-2082/1–390761711). This research received support from the Elizabeth Blackwell Institute for Health Research and the EPSRC Impact Acceleration Account EP/R511663/1 University of Bristol to C.S. and I.B., the Deutsche Forschungsgemeinschaft via the Gottfried-Wilhelm-Leibniz Program to H.D. and J.P.S., a European Research Council Consolidator Grant (#724261) to H.D., the Max Planck School Matter to Life (a joint program of BMBF and Max Planck Society) to O.S., M.M., H.D., and J.P.S., and the EU FET Open Project Virofight (#899619) to H.D. I.B. acknowledges support from UK Research and Innovation (UKRI) through the Bristol Synthetic Biology Centre BrisSynBio (BB/L01386X/1). C.S. and I.B. are Investigators of the Wellcome Trust (210701/Z/18/Z; 106115/Z/14/Z).

## Author contributions

O.S., I.P., I.B., and J.P.S. conceived and designed the study. K.G. cloned, expressed, and purified S proteins aided by K.V. G.S. performed ribosome display experiments and selected ADAH11. G.S. and K.G. produced and purified ADAH11. J.E.H.B. performed QCM-D measurements. F.K., C.Si., and H.D. acquired and analyzed cryo-EM data. S.F. established MS protocols and analyzed MS data. M.M. performed nanoparticle tracking analysis. E.A.C.A. conceptualized integrin blocking experiments. C.S. helped to conceive and design the study. A.R. designed and performed SARS-CoV-2 competition assays. O.S. performed microscopy observations, DLS measurements, retention assays, and ASA analysis. O.S., I.P., and J.P.S. wrote the manuscript with input from all authors.

## Funding

## Competing interests

C.B.S. and I.B. declare shareholding in Halo Therapeutics Ltd, related to this Correspondence. Patents describing materials, material compositions and formulations related to the present findings have been filed by O.S., K.G., I.P., J.P.S., C.B.S., and I.B. The remaining authors declare no competing interests.
