## [Peer Review File · Nature Communications]

Reviewers' Comments:

Reviewer #1:

Remarks to the Author:

The authors developed a minimal virion (MiniV) of SARS-CoV-2 that mimics the structure and receptor binding of natural SARS-CoV-2 virion via bottom-up assembly. Using this MiniV system, the authors show that a highly conserved free fatty acid binding pocket (FABP) with SARS-CoV-2 S protein regulates the S-mediated cell binding via binding of FFAs and mediating the S open to-closed equilibrium. The authors also investigate the physiological impact and therapeutic potentials of FABP. In summary, this work provides an in-depth understanding of S protein FABP, and its impacts in SARS-CoV-2 infection. This reviewer has the following suggestions to improve the paper.

Specific comments:

1. In Figure 2a, whether the concentrations of PUFAs used to incubate with ApoS MiniV affected the final binding efficiency. How were concentrations determined, and concentrations within the biological realm? Likewise, it is not clear what the concentrations of compounds used in Figure 2f. Moreover, although Dex and Vitam K2 treatment significantly reduced the MiniV binding to cell, the difference in the mean decrease between the control and compound-treated group was small (less 10%, from 100% decrease to ~92%). Examining the EC50 values of these compounds may be more indicative of their clinical potentials.
2. In Figure 2a and 2e, why the addition of poly-unsaturated FFAs (PUFAs), OA, LA and AA reduced MiniV binding to cell but enhanced the binding efficiency of MiniVs blocked with linRGD?
3. Whether FFA binding affects the enzymatic activity of host TMPRSS2 or cathepsin L that is critical for binding and entry process of SARS-CoV-2.
4. The authors only examined the binding efficiency of MiniV with different treatments at 24 h post incubation throughout almost all the study. A time-course assay is required for better understanding the dynamic change of MiniV binding to cell in the presence of different PUFAs.
5. It is better for the authors to use authentic SARS-CoV-2 to test the impact of LA/AA concentrations on the neutralization activity of patient serum IgGs.

Reviewer #2:

Remarks to the Author:

This excellent paper by Stauer et al., describes the development and evaluation of synthetic variations as tools for studying SARS-CoV2. This paper holds great importance as we observe SARS-CoV2 mutate, and in the need to develop updated vaccines.

While the paper is of great interest, the authors' choice to work with MCF7 breast cancer cells alone is extremely puzzling.

One would expect to see some of this data with cell lines (or even in vivo) of the respiratory system, and cells of other organs that are affected by SARS-CoV2.

Once such work is added this paper will make a big splash in the synthetic biology and vaccine world.

Point-by-point response:

Reviewer #1:

The authors developed a minimal virion (MiniV) of SARS-CoV-2 that mimics the structure and receptor binding of natural SARS-CoV-2 virion via bottom-up assembly. Using this MiniV system, the authors show that a highly conserved free fatty acid binding pocket (FABP) with SARS-CoV-2 S protein regulates the S-mediated cell binding via binding of FFAs and mediating the S open to-closed equilibrium. The authors also investigate the physiological impact and therapeutic potentials of FABP. In summary, this work provides an in-depth understanding of S protein FABP, and its impacts in SARS-CoV-2 infection. This reviewer has the following suggestions to improve the paper.

We thank the reviewer for considering our work of importance for understanding the FABP function in the SARS-CoV-2 spike protein.

Point 1: *In Figure 2a, whether the concentrations of PUFAs used to incubate with ApoS MiniV affected the final binding efficiency. How were concentrations determined, and concentrations within the biological realm?*

We thank the reviewer for pointing out this missing piece of information. An important advantage of our MiniV system is that it is quantitatively defined. We can therefore control the number of spike proteins added to the cultures and therefore stoichiometrically adjust the amount of FFAs required to saturate the FABP. We previously reported the affinity of S for FFAs is considerably high with a K_D for LA of 26 nM (see Toelzer et al. (2020) *Science*). For **Figure 2a**, we therefore added FFAs in a 2x molar excess to the MiniV solution to saturate the FABP. As we chose physiologically relevant MiniV particle concentrations (see page 6, line 6), a final concentration of 1 μ M FFAs was added.

As described on page 16, lines 8-11 of our original manuscript, under non-inflammatory conditions individual FA levels are kept below 0.1 μ M (Brash et al. (2001) *The Journal of Clinical Investigation*; Richieri et al. (1995) *Journal of Lipid Research*). However, in COVID19 patients FA level raise to 1 μ M and above (Yan et al. (2019) *Viruses*; Archambault et al. (2020) *medRxiv* doi:10.1101/2020.12.04.20242115). Therefore, and importantly for the design of our study, the FFAs concentration applied are within the relevant physiological range.

Based on the reviewers' comment, we now realized that our manuscript will benefit from a more detailed description of FFA concentrations added to the MiniV solution, apart from the experimental description in the materials and methods section. In the revised version of our manuscript, we now clarify in the main text on page 9, line 23 to page 10, line 1:

"MiniVs were loaded with 2x molar excess of FFAs with respect to FABPs in S, equaling 1 μ M, which is comparable to the FA levels in sera of COVID19 patients^{26,27}. Based on the nanomolar affinity of S for FFAs¹, this concentration is expected to result in complete saturation of all binding pockets as partially supported by our mass spectrometry quantification (see below)."

Further to Point 1: Likewise, it is not clear what the concentrations of compounds used in Figure 2f. Moreover, although Dex and Vitam K2 treatment significantly reduced the MiniV binding to cell, the difference in the mean decrease between the control and compound-treated group was small (less 10%, from 100% decrease to ~92%). Examining the EC50 values of these compounds may be more indicative of their clinical potentials.

A similar rationale as described for the FFAs above was applied for the experiments shown in Figure 2f. Consequently, results shown in Figure 2f were obtained with a final drug concentration of 1 μ M. However, in contrast to FFAs, the specific K_D values of the drugs for the FABP are not known. Therefore, the data presented in Figure 2f are “drug-normalized” retention values. This normalization procedure takes several controls into account (e.g. the retention of naïve SUVs under drug treatment and the retention of ApoS MiniVs), as described in the corresponding materials and methods section. The normalization was performed in order to account for any non-FABP related effects in the cellular behaviour induced by drug treatment (e.g. changes in cellular morphology or homeostasis) that might affect MiniV binding.

However, based on the reviewers’ comment, we acknowledge that this might be a precise experimental assessment but also a rather theoretical and partially non-intuitive measurement. For the reader, this might complicate intuitive understanding of the experimental procedure and the conclusions drawn. In the revised version of our manuscript, we therefore now also report the IC50 values for Dexamethasone and Vitamin K2 measured in lung epithelial cells (see page 12, lines 13-15). We agree that this will contribute to the comprehensibility of the experimental findings. We now also indicate the final drug concentration of 1 μ M applied in Figure 2f in the main text (page 12, line 11) and the figure legend.

Point 2: In Figure 2a and 2e, why the addition of poly-unsaturated FFAs (PUFAs), OA, LA and AA reduced MiniV binding to cell but enhanced the binding efficiency of MiniVs blocked with linRGD?

We thank the reviewer for pointing out this potentially partially misleading description of our data representation in our original manuscript. Figure 2a shows the results of a retention analysis, where MiniV fluorescence intensity in treated cultures is compared before and after washing. Figure 2e, however, shows a linRGD-normalized retention analysis calculated as (described in our materials and methods section):

$$\frac{R - R_{linRGD}}{R_{nativeS}}$$

Accordingly, the y-axis in this graph indicates a normalized difference. Figure 2a reports an absolute retention value. This was done in order to relate the difference between control and linRGD-treated cultures to the standard treatment control of MiniVs with native S to account for linRGD-mediated changes in cell behavior (similarly to the drug-normalization performed in Figure 2f).

Based on the reviewers’ comment, we now understand that this normalization step is not highlighted enough in the main text description of the figure and might be misleading to the

reader. Therefore, we now introduced the following explanation into the main text of our revised manuscript (page 11 line 24 to page 12 line 1):

“For this, we measured the native S-normalized differences in retention between linRGD-blocked cultures and control cultures incubated with MiniV that were loaded with FAs (see methods for normalization of retention).”

Point 3: Whether FFA binding affects the enzymatic activity of host TMPRSS2 or cathepsin L that is critical for binding and entry process of SARS-CoV-2

We agree with the reviewer that the question of whether cell-mediated post-translational processing of S is affected by the FABP is of high interest. Proteases like TMPRSS2 cleave S within the S2 region of the protein. According to our previously published cryoEM structure of S (Toelzer *et al.* (2020) *Science*) the FABP resides within the S1 domain and mostly affect initial binding events. However, we also report in a recent pre-print (Oliveira *et al.* (2021) *bioRxiv* DOI:10.1101/2021.06.07.447341) results on dynamical-nonequilibrium molecular dynamics simulations that the FABP is coupled to functionally relevant region far away from the RBD. Our MiniV system is perfectly suited to tackle this question experimentally.

We therefore performed TMPRSS2 blocking experiments by pre-incubation of the cells with camostat mesylate, a known inhibitor of this protease (Hoffmann *et al.* (2021) *The Lancet EBioMedicine*) and measured MiniV retention of ApoS and FA-loaded S MiniVs. Our new results were added as **Figure S8** to the revised version of our manuscript and discussed on page 10, line 18 to page 11 line 1. Interestingly, we could not measure significant differences in MiniV retention when blocking protease cleavage at several different camostat mesylate concentrations (200 μ M, 20 μ M and 2 μ M), although the recombinant S applied in our study does contain the S2' cleavage site. We could also not find a significant effect when the FABP was loaded with FAs. These data imply that the FABP is most crucial for regulation of initial cell binding rather than for S post-translation modification, fusogenic transformation and cellular uptake.

Figure S8.

Protease-based S processing. Retention assays for SUVs and MiniVs incubated with A549 human alveolar basal epithelial cells for 8 hours. Retention assays were performed either under untreated control conditions or under treatment with the TMPRSS2 inhibitor camostat mesylate (CM). MiniVs were produced presenting native S or ApoS on the surface. Results are shown as mean \pm SD from 3 biological replicates, n.s. = not significant, unpaired two-tailed t-test.

Point 4: *The authors only examined the binding efficiency of MiniV with different treatments at 24 h post incubation throughout almost all the study. A time-course assay is required for better understanding the dynamic change of MiniV binding to cell in the presence of different PUFAs.*

We agree with the reviewer that the time dynamics of FABP-regulated MiniV-cell binding is of high interest. Additionally, to our initial time-resolved analysis presented in **Figure 1g**, we now present in **Figure 1h and 1i** experimental analysis resolving the time-dynamics of MiniV retention in human alveolar basal epithelial cells and human umbilical vein endothelial cells (see also Reviewer #2). Moreover, we performed time dynamic analysis of retention also for ApoS and FFA-loaded MiniVs measured with human alveolar basal epithelial cells. In accordance to our initially reported endpoint analysis, we observed increased retention of ApoS MiniVs while FFA-loaded MiniVs showed reduced retention. These results have been included as **Figure S6** in the revised version of our manuscript.

Figure S6.

Time resolved retention assays for FFA loading. Time resolved retention assays of MiniVs presenting native S, ApoS or LA and AA-loaded ApoS and incubated with A549 human alveolar basal epithelial cells for 24 hours. Results are shown as mean \pm SD from 3 biological replicates, * p <0.05, n.s. = not significant, unpaired two-tailed t-test.

Point 5: It is better for the authors to use authentic SARS-CoV-2 to test the impact of LA/AA concentrations on the neutralization activity of patient serum IgGs.

The reviewer raises an interesting point concerning the translational aspects of our MiniV system. We agree that after reaching new insights with our well-controlled MiniVs, the effects of FFAs on neutralization should be verified in a live virus system. We therefore performed live virus neutralization assays with COVID-19 convalescent human IgGs in human alveolar basal epithelial cells under high and low FFA conditions. We found that the IC₅₀ of viral neutralization of the IgGs was increased approximately one order of magnitude under high FFA conditions (corresponding to the FFA concentration under inflammatory COVID-19 conditions). This demonstrates that findings obtained with the well-controlled MiniV system can be translated to live viruses, although this system is less-well controlled and it cannot be finally resolved if the FFAs affect the viral or the cellular biology in this setting. We now included these results as **Figure S15** into our revised manuscript and discuss the findings on page 16, lines 16-19.

Figure S15.

Live virus neutralization assays. Serial dilutions of IgG antibodies were pre-incubated with SARS-CoV-2 natural viruses for 1 h before infection of ACE2-expressing A549 human alveolar basal epithelial cells. Eight hours post-infections, ORF7a mRNA expression levels were quantified by qRT-PCR and normalized to infection levels in absence of IgG antibodies (non-treated). Results are shown as mean \pm SD from 3 independent experiments. Nonlinear regression is indicated by the purple line. Relative inhibitory concentration of 50% (IC₅₀) values is indicated by the dotted line.

Reviewer #2:

This excellent paper by Staufer et al., describes the development and evaluation of synthetic variants as tools for studying SARS-CoV2. This paper holds great importance as we observe SARS-CoV2 mutate, and in the need to develop updated vaccines.

We thank the reviewer for considering our work of importance for the continuous improvement of vaccination strategies and our understanding of SARS-CoV-2.

Point 1: *While the paper is of great interest, the authors' choice to work with MCF7 breast cancer cells alone is extremely puzzling. One would expect to see some of this data with cell lines (or even in vivo) of the respiratory system, and cells of other organs that are affected by SARS-CoV2. Once such work is added this paper will make a big splash in the synthetic biology and vaccine world.*

We thank the reviewers for raising this point and agree that additional experimental verification with cell types of COVID-19 diseased tissues would be beneficial to our manuscript and increase the scope for physiological relevance. We therefore reproduced our key findings with human alveolar basal epithelial cells, one of the primary target cell types of SARS-CoV-2. To further increase the physiological relevant scope, we additionally measured MiniV retention to primary human umbilical vein endothelial cells. Next to lung tissue, endothelial cells of the blood vessel system are majorly affected in progressed COVID-19 patients and key contributors to the severe disease states (Teuwen *et al.* (2020) *Nature Reviews Immunology*).

We performed several new experiments with these cell types (that were partially requested by Reviewer #1) and have now revised our main **Figures 1 and 2** accordingly. Moreover, we now added **Figures S3, S6, S8, S14 and S15** to the revised version of our manuscript presenting new experimental findings with the lung epithelial and endothelial cells (see revised figures attached below). Particularly, we performed time course analysis of MiniV binding to these cells (revised **Figure 1g,i**), assessed the influence of FFA loading in the FABP on MiniV binding (revised **Figure 2b,c**), infected human alveolar basal epithelial cells with live SARS-CoV-2 virus to validate the effects of FFAs on IgG neutralization obtained with the MiniV system (**Figure S15**), assessed the effect of TMPRSS2 cleavage in FFA-loaded S MiniVs (**Figure S8**) as well as assessed neutralization of ADAH11 in dependence of FABP occupancy (**Figure 14**). In both cell types, we were able to reproduce our key findings obtained with the MCF7 cell line. Intriguingly, we observed significantly intracellular uptake of the MiniVs in these cells as compared to the MCF-7 cell system. An observation that we now describe in the main text of our manuscript (see page 6 lines 9-10) and report in confocal microscopy images in **Figure S3**. We thank the reviewer, as based on his/her suggestion we were able to expand the physiological relevance of our findings, improving our manuscript.

Figure 1. Bottom-up assembly of minimal SARS-CoV-2 virions. (a) Schematic illustration of MiniVs based on SUVs with SARS-CoV-2 S ectodomains, immobilized via their His-tag. **(b)** Lipid formulation of SUVs derived from the ERGIC

with NTA-functionalized and fluorescent lipids. **(c)** MiniV and SUV size distribution analysis by dynamic light scattering. **(d)** Exemplary cryo-EM tomography slices of MiniVs with immobilized S on the membrane. Scale bar is 50 nm. **(e)** Representative confocal microscopy images of MCF7 human epithelial cells incubated for 10 min with MiniVs, showing attachment of the MiniVs to the cells surface. Inset shows magnified area of attachment. Scale bar is 7 μ m. **(f)** Maximal confocal microscopy z-projections of MCF7 human epithelial cells incubated for 18 hours with MiniVs (top row) or with SUVs lacking S on the surface (bottom row). Scale bar is 40 μ m. **(g,h,i)** Time resolved retention assay of MiniVs and SUVs incubated with MCF7 (g), A549 (h) and HUVEC (i) cells. **(j)** SUV-normalized retention assay for MiniVs presenting different recombinant hCoV S variants incubated for 24 h with MCF7 cells. **(k)** Retention assay for MiniVs presenting SARS-CoV-2 D614G and B.1.1.7 S variants incubated for 24 h with MCF7 human epithelial cells. Results are shown as mean \pm SD from at least 3 biological replicates, *p<0.05, **p<0.005, unpaired two-tailed t-test.

Figure 2. FABP-based regulation of S binding. (a,b,c) Native S-normalized retention assay for MiniVs presenting ApoS, loaded with 1 μ M FFAs, after incubation with MCF7 cells (a), A549 human alveolar basal epithelial cells (b) and human umbilical vein endothelial cells (c). (d) Molecular surface representation of the S trimer cryo-EM

structure (PDB 7BNN) in top view with one open RBD exposing the RGD motif (orange). ACE2 binding residues in the receptor binding motif are shown in purple. (e) Cartoon structure representation (PDB 6ZB5) of the LA-bound FABP (green), the acidic headgroup anchor (blue) and the adjacent RGD motif (orange) in the LA-locked S conformation. (f) SUV-normalized retention assay for MiniVs presenting native S, R403A mutated S without RGD motif or native S incubated with 20 μ M linRGD for integrin blocking. Retention was measured after 24 h incubation with MCF7 cells. (g) LinRGD-normalized retention assays for MiniVs presenting FFA-loaded ApoS after 24 h incubation with MCF7 cells. (h) Drug-normalized retention assay for ApoS-MiniVs incubated with MCF7 cells and 1 μ M of potential S binding drugs. Results are shown as mean \pm SD from at least 3 biological replicates, * p <0.05, ** p <0.005, n.s.=not significant, unpaired two-tailed t-test.

Figure S3. Microscopy analysis of MiniV uptake. Representative confocal microscopy images of MiniV binding and uptake to human alveolar basal epithelial cells (top) and human umbilical vein

endothelial cells (bottom) after incubated for 8 hours with Scale bars are 30 μm (top) and 10 μm (bottom).

Figure S6.

Time resolved retention assays for FFA loading. Time resolved retention assays of MiniVs presenting native S, ApoS or LA and AA-loaded ApoS and incubated with A549 human alveolar basal epithelial cells for 24 hours. Results are shown as mean \pm SD from 3 biological replicates, * $p < 0.05$, n.s. = not significant, unpaired two-tailed t-test.

Figure S8.

Protease-based S processing. Retention assays for SUVs and MiniVs incubated with A549 human alveolar basal epithelial cells for 8 hours. Retention assays were performed either under untreated control conditions or under treatment with the TMPRSS2 inhibitor camostat mesylate

(CM). MiniVs were produced presenting native S or ApoS on the surface. Results are shown as mean \pm SD from 3 biological replicates, n.s. = not significant, unpaired two-tailed t-test.

Figure S14.

FFA-regulated neutralization potency of IDNS-directed nanobody. **(a)** Native S-normalized neutralization of ADAH11 for MiniVs presenting FFA-loaded ApoS incubated with A549 human alveolar basal epithelial cells for 8 hours. **(b)** Native S-normalized neutralization of ADAH11 for MiniVs presenting FFA-loaded ApoS incubated with human umbilical vein endothelial cells for 8 hours. Results are shown as mean \pm SD from 3 biological replicates, * p <0.05, n.s. = not significant, unpaired two-tailed t-test.

Figure S15.

Live virus neutralization assays. Serial dilutions of IgG antibodies were pre-incubated with SARS-CoV-2 natural viruses for 1 h before infection of ACE2-expressing A549 human alveolar basal epithelial cells. Eight hours post-infections, ORF7a mRNA expression levels were quantified by qRT-PCR and normalized to infection levels in absence of IgG antibodies (non-treated). Results

are shown as mean \pm SD from 3 independent experiments. Nonlinear regression is indicated by the purple line. Relative inhibitory concentration of 50% (IC₅₀) values is indicated by the dotted line.

Reviewers' Comments:

Reviewer #1:

Remarks to the Author:

The authors have submitted a thoroughly revised version of their manuscript, in which all critical points have been addressed. They have provided more convincing evidence to strengthen their conclusions. This reviewer has no further comments.

Reviewer #1

Point 1: The authors have submitted a thoroughly revised version of their manuscript, in which all critical points have been addressed. They have provided more convincing evidence to strengthen their conclusions. This reviewer has no further comments.

We thank the reviewer for his/her critical and constructive comments to our manuscript.